# Inflammatory Status and Glycemic Control Level of Patients with Type 2 Diabetes and Periodontitis: A Randomized Clinical Trial

**DOI:** 10.3390/ijerph18063018

**Published:** 2021-03-15

**Authors:** Biagio Rapone, Elisabetta Ferrara, Massimo Corsalini, Erda Qorri, Ilaria Converti, Felice Lorusso, Maurizio Delvecchio, Antonio Gnoni, Salvatore Scacco, Antonio Scarano

**Affiliations:** 1Department of Basic Medical Sciences, Neurosciences and Sense Organs, “Aldo Moro” University of Bari, 70121 Bari, Italy; gnoniantonio@gmail.com (A.G.); salvatore.scacco@uniba.it (S.S.); 2Complex Operative Unit of Odontostomatology, Hospital S.S. Annunziata, 66100 Chieti, Italy; igieneeprevenzione@gmail.com; 3Interdisciplinary Department of Medicine, University of Bari, 70121 Bari, Italy; massimo.corsalini@uniba.it; 4Dean Faculty of Medical Sciences, Albanian University, Bulevardi Zogu I, 1001 Tirana, Albania; erda79@yahoo.com; 5Department of Emergency and Organ Transplantation, Division of Plastic and Reconstructive Surgery, “Aldo Moro” University of Bari, 70121 Bari, Italy; ilaria.converti@gmail.com; 6Department of Oral Science, Nano and Biotechnology and CeSi-Met University of Chieti-Pescara, 66100 Chieti, Italy; drlorussofelice@gmail.com (F.L.); ascarano@unich.it (A.S.); 7Department of Metabolic and Genetic Diseases, Giovanni XXIII Children’s Hospital, 70126 Bari, Italy; mdelvecchio75@gmail.com

**Keywords:** type 2 diabetes, periodontitis, periodontal inflammation, systemic inflammation, C-reactive protein, non-surgical periodontal treatment, dental public health

## Abstract

Background: Based on the holistic approach to prevention diabetic disease, the role of periodontal inflammation in type 2 diabetes mellitus (T2DM) is under intensive scrutiny. Data from clinical trials have shown benefit from a periodontal therapy in providing patients with type 2 diabetes improvement despite relatively disappointing long-terms response rates. The aim of this study was to investigate the short-term glycemic control level and systemic inflammatory status after periodontal therapy. Methods: This was a randomized trial with a 6-months follow-up. Participants aged 56.4 ± 7.9 years with diagnosed type 2 diabetes and periodontitis were enrolled. Among the 187 type 2 diabetic patients, 93 were randomly assigned to receive non-surgical periodontal treatment immediately and 94 to receive the delayed treatment. Within and between groups comparison was done during the study period, and the differences between groups were assessed. Results: The difference between HbA1c values at baseline (*Mdn* = 7.7) and 6 months after non-surgical periodontal treatment (*Mdn* = 7.2) was statistically significant, *U* = 3174.5, *p* = 0.012, *r* = 0.187. However, although technically a positive correlation, the relationship between the glycated hemoglobin value and periodontal variables was weak. The differences between both the groups over 6 months were not statistically considerable, failing to reach statistical significance. At 6 months the difference between groups about the C-reactive protein (CRP) levels was statistically significant, *U*=1839.5, *p* = 0, *r* = 0.472, with a lower concentration for the intervention group. Furthermore, the intervention group showed a statistically significant difference between baseline and 6 months evaluation (*U* = 2606.5, *p* = 0, *r* = 0.308). Conclusions: The periodontal intervention potentially may allow individuals with type 2 diabetes to improve glycemic control and CRP concentrations, and diabetes alters the periodontal status.

## 1. Introduction

Despite advances in prevention and treatment measures, diabetes mellitus remains one of the most important chronic diseases worldwide, associated with high mortality and morbidity, with an estimated 300 million cases in the year 2025 [1]. Research has pointed to the multiplicative effect when several risk factors are present and, of interest, scientific evidence underlines the possibility of closely linked inflammatory etiology. Accumulating evidence supports a key role of periodontal inflammation in the pathophysiology of several systemic disorders [2,3,4]. Progress in understanding the pathogenesis of diabetes has generated increasing interest in targeting inflammatory pathways and biomarkers of inflammation to help prevent and control diabetes and related sequelae [5,6,7,8]. Periodontitis is a chronic multifactorial inflammatory disease affecting the tissues surrounding the teeth [7], and has a prevalence estimated at 20–50% of the global population. Commonly, over 10% of the older population has severe disease [8,9,10]. Periodontitis can promote systemic chronic inflammation (SCI) that can, in turn, lead to exacerbation of type 2 diabetes [9,10,11]. This consistent clinical evidence for an interconnection between periodontitis and diabetes risk comes from multiple randomized controlled trials (RCTs) and epidemiological investigations data that have proven that periodontal treatment resulted in HbA1c levels reduction [10,12,13]. All these findings are in harmony with the hypothesis that periodontitis and diabetes may fuel each other in a bidirectional pathway [9,13], in which a reciprocal action of inflammatory mediators leads to worsen of pathological conditions, becoming closely linked diseases [11].

The metabolic dysregulation in diabetes associated with hyperglycemia and hyperinsulinemia are predisposing factors to infectious diseases in diabetes. Periodontitis is one of the most common complications of diabetes mellitus, affecting more than 50% of the diabetic population [14]. Periodontitis has been implicated in an increased risk of type 2 diabetes, and diabetes patients are two to five times as likely to develop risk of suffering periodontitis compared to healthy individuals, but the molecular mechanisms linking these two diseases are partially known [5,15,16,17,18,19]. Mechanistic studies in in vivo conditions agree to assert that among regulatory factors of this connection should be explored [20]. Specifically, the pathway model that aggregates the knowledge about the impact of diabetes on periodontal status have been well-documented and characterized [14]. Data on the clinical impact of diabetes on periodontal inflammation enhancement come from two sources: the potential direct effect of hyperglycemic status, which may be responsible for the oral microbiome shifting, impaired cellular function and host defense, and upregulation of the circulant proinflammatory mediators known to be released primarily by immune cells [21,22,23,24,25,26,27]; the accelerated glycation of protein and lipids mediated by Advanced glycation end products (AGES) and fatty acids, which may indirectly contribute to the degenerative process of periodontal tissues and influence negatively osteogenesis [28,29,30,31,32,33]. In contrast, it has been assumed that long term low-level exposure to periodontal inflammation have a negative impact on metabolic control, but the pathway by which periodontitis contributes to worsening diabetes is still unclear [16]. Human studies have demonstrated that patients suffering from periodontitis exhibit higher concentrations of proinflammatory cytokines, such as tumor necrosis factor-alpha (TNF-α), C-reactive protein (CRP), interleukin-6 (IL-6) and Il-1β. Several mechanisms of altered macrophage polarization during obesity have been recently suggested. Based on this evidence, dysregulation in the cytokine network may contribute to development or sustaining diabetes via activation of the chronic systemic inflammation [14,15,16,21]. The clinical relevance of increases of CRP in subjects with periodontitis has been demonstrated by showing that the peripheral inflammatory response is reflected in increased concentrations of the C-reactive protein in patients with periodontal infections compared to periodontally health individuals. The most reliable evidence for the co-occurrence of diabetes and periodontitis and the correlation between the two diseases is a function of inflammatory pathway [2,5,13,30].

Based on the simultaneous occurrence of periodontitis and diabetes, as recorded in most of clinical trials, this paper briefly resumes the conceptual cycle that accounts for the theory of bidirectional relationship between diabetes and periodontitis and explores the impact of suppression of periodontal inflammatory response pathway in minimizing the interrelation between the peripheral and systemic inflammatory status. In this perspective, the chief object of this trial was aimed to investigate the effectiveness of non-surgical periodontal treatment on glycemic control in patients diagnosed type 2 diabetes and, simultaneously, the potential improvement of systemic inflammatory status.

## 2. Materials and Methods

### 2.1. Study Design and Participants

The study was a single center double-blind with two parallel-groups, randomized controlled trial with a follow-up of 6 months. The trial was conducted in compliance with the provisions of the Declaration of Helsinki and Good Clinical Practice guidelines. The study was reviewed and received ethical approval from the Institutional Review Board of Albanian University, Number 385. Patients were recruited between June 2018 and January 2020 and included if diagnosed with type 2 diabetes and if their diabetes therapy had remained unchanged over the previous 3 months and they had not participated in DM educational programs prior to the commencement of study. Further, to be included, patients were required to have a diagnosis of periodontitis [20].

### 2.2. Recruitment, Randomization and Blinding Status

Potentially eligible patients were recruited based on medical record data. Patients with a diagnosis of type 2 diabetes, within the previous 3 years, and having diagnosis of periodontitis [23]. Diagnostic criteria of periodontitis have been validated since 1999 [27] by the American Academy of Periodontology (AAP) and have been enlarged in the 2017 revision [28]. According to the American Academy of Periodontology (AAP), clinical diagnosis of periodontitis was made if clinical attachment loss (CAL) affected ≥ 2 non-adjacent teeth or buccal/oral CAL of ≥ 3 mm with pocketing of > 3 mm was detectable at ≥ 2 teeth, and the detected CAL cannot be attributed to traumatic causes, dental caries, endodontic lesion or molar malpositions. Patients were excluded if they: had insulin dependent diabetes mellitus, or higher chronic disease, smoking or consuming alcohol, had used any antibiotics or long-term anti-inflammatory drugs in the last 6 months prior to the trial; females who are pregnant, lactating or less than 6 weeks post-partum; having received periodontal treatment within the previous year, were unable to provide informed consent or comply with study requirements at the time of the recruitment encounter and were younger than 18 years. After the trial eligibility criteria were confirmed, each patient provided informed consent, which was mandatory before randomization. Randomization was done using a computer-generated series of numbers that ensured balance between experimental groups. Prior to conducting the research study, sample size was determined, setting *Type I error (α)* at 0.05 and *Type* II *error* (β) at 0.02 and power at 80%. Participants were randomly assigned, in a 1:1 ratio, to receive either the non-surgical periodontal treatment (intervention group—IG) or delayed non-surgical periodontal treatment (control group —CG). Due to the nature of the intervention neither participants were masked to allocation, while the trial statistician and one clinical trial personnel were blinded to the study groups during data analysis. The design and methods of this RCT were in accordance with the recently published extension of the CONSORT statement to randomized trials of non-pharmacological treatment. Figure 1 illustrates the design of the study in the form of a CONSORT diagram.

### 2.3. Assessment of Clinical Periodontal Parameters

At baseline and at 3 and 6 months after non-surgical periodontal treatment, the following clinical parameters were detected to record gingival and periodontal status by a single examiner using a standardized manual periodontal probe (CP11 Hu Friedy, Europe) in six sites of each tooth (distobuccal surface, centre of vestibular surface, mesio-buccal surface, distolingual surface, centre of lingual surface and mesiolingual surface): pocket depth (PD) [1], to assess the periodontal status by measuring the space between the pathologically detached gingiva and the tooth; clinical attachment level (CAL), to assess the periodontal status by measuring the distance from the cemento–enamel junction of the tooth to the pocket base; gingival index (GI-Löe19), to assess the severity of gingival inflammation on the basis of chromatic evaluation, consistency and bleeding on probing; plaque index (PI—Silness and Löe), to assess the thickness of bacterial plaque at the cervical margin of the teeth. In order to reduce the effects of the examination variability, the technique proposed by Silness and Löe was used on determining the PI score [2], mesial (M) and distal (D) measurements. Glycated hemoglobin (HbA1c) and *CRP* were measured at baseline and during the third and sixth month of follow-up at a local diabetic center.

### 2.4. Non-Surgical Periodontal Treatment

The practical application of periodontal treatment is the result of the ambivalence of periodontal disease, consisting of biologic etiological components and a psychological issue stood regarding the perception of individuals need to address the pathological condition. Then, the earliest stage of periodontal treatment for the case group consisted of motivational intervention helping patients change behavior associated with customized oral hygiene instructions for daily plaque control, pointed out on the potential successful outcomes, including the improvement of periodontal disease and potential benefit for diabetes control. At the end of the first phase, active professional treatment was carried out. Scaling and root planing were performed under local anesthesia and treatment was divided in four sessions within 24 h, each of 45 min for quadrant and were performed using an ultrasonic scaler (PIEZO-soft ultrasonic scaler; KaVo Dental, Germany) equipped with PIEZO Scaler Tip 201, and manual instruments (Gracey Curettes, SG 1/2, 3/4, 5/6, 7/8, 11/12, 13/14; Gracey Curette, SAS 3/4, 11/12, 13/14; Hu-Friedy, USA). The ultrasonic instrumentation was performed in apical to cervical direction using linear oscillations and at 30 KHz with a medium power setting [3]. A continuous water spray on the working area was maintained, and a subgingival irrigation with an antiseptic mouthwash (chlorhexidine 0.12%) was associated. All patients were treated by the same operators using the standard curettes angle and force applications. At the end of the 6-month follow-up period the non-surgical periodontal treatment was carried out for the control group, performing the same procedures for an ethical reason.

### 2.5. Study Outcomes

The primary outcome of the clinical trial was the change from baseline to 6 months endpoint in glycosylated hemoglobin (HbA1c) (time frame: baseline, 3 months, 6 months). The secondary outcomes were the change of periodontal measurements. Degree was categorized, based on primary criteria and the presence of risk factors as Grade A, characterized by a slow rate of progression; Grade B, characterized by a moderate rate of progression and Grade C, characterized by a rapid rate of progression. Severity was determined based on CAL, and categorized as: Stage I: periodontitis, with a mean of 1 mm of periodontal attachment loss; Stage II: moderate periodontitis, with a mean of 3 mm of periodontal attachment loss; Stage III: severe periodontitis, with ≥5 mm of periodontal attachment loss, and teeth loss ≤4 and Stage IV: advanced periodontitis, with ≥5 mm of periodontal attachment loss teeth loss ≥5.

### 2.6. Governance and Ethics

The study was approved by the local Ethics Committee of Albania University, Tiran, Albania (Nr. 385 Prot.). The study was conducted according to the guidelines of the Declaration of Helsinki. Informed consent was obtained from all subjects involved in the study.

### 2.7. Statistics 

Statistical analyses were conducted by using SPSS software (Statistical Package for the Social Sciences, version 14.0, SPSS Inc., Chicago, IL, USA). The alpha level was set at 5%. Normality was verified by applying the Shapiro–Wilk test. Examined variables did not present normal distribution, and consequently, a non-parametric approach was used. Additionally, box plots were visually scrutinized, and descriptive analysis, skewness and kurtosis were used to evaluate deviation from normality. A *p*-value of >0.05 indicated a normal distribution. To compare the variables within- and between-groups, the Mann–Whitney test was applied at each time-point. Correlation was analyzed using Spearman’s correlation coefficient. Multiple linear regression analysis was performed to assess relative association between HbA1C and periodontal parameters assuming HbA1C the dependent variable and periodontal indices the independent variables. 

Our sample size calculation was determined to detect a difference in change in the primary outcome, HbA1c of 0.5% between groups from baseline. Based on the assumption of a standard deviation of 2 mmol/mol (0.1%) with a 5% significance level using a two-sided two-sample z-test, and a drop-out rate of 20%, minimum of 68 participants per group would give us 80% to detect a difference in HbA1C of 5 mmol/mol (0.5%) between the two study groups. Participants’ characteristics were described with mean and standard deviations for normally distributed data. Data with a skewed distribution are presented as medians with an interquartile range. Statistical significance was inferred at a two-tailed *p* value of <0.05.

## 3. Results

A total of 265 subjects were screened, and 187 subjects were randomized into the study. The population consisted of 81 male (45%) and 106 females (56%) in total, with a mean age of 56.4 ± 7.9. As shown in Figure 1, the 6-month trial was completed by 90 of the 187 participants (48%) in the intervention group (IG) (four refused treatment) and by 90 of the participants (48%) in the control group (CG) (three refused to continue participation). 

Table 1 shows the baseline participants characteristics. 

The descriptive analysis of clinical parameters of both groups at baseline is reported in Table 2.

### 3.1. Correlation Analysis between the Variables of Intervention Group at Baseline

A Spearman rank correlation was performed to test if there was a relationship between the HbA1c level and periodontal parameters. At baseline, results indicated that there was no significant association between HbA1c and PI, *r* (88) = 0.106, *p* = 0.322, and for GI: *r* (88) = 0.126, *p* = 0.237. As shown in Table 3, no significant association between HbA1c and PD was found, *r* (88) = −0.136, *p* = 0.201 and no statistically significant association was revealed between HbA1c and CAL: *r* (88) = −0.142, *p* = 0.181. A significant positive association between PD and CAL in the intervention group, *r* (88) = 0.718, *p* = 0 was found. The results of the Spearman rank correlation indicated that there was a positive association between the CRP value and HbA1c and periodontal parameters, but no significant association was revealed (PI, *r* (56) = −0.224, *p* = 0.091; GI, *r* (56) = −0.205, *p* = 0.123; PD, *r* (56) = −0.224, *p* = 0.091; CAL, *r* (56) = −0.01, *p* = 0.938; HbA1c *r* (56) = −0.222, *p* = 0.093).

At 3 months (Table 4), no significant association between HbA1c and GI was found, with *r* (88) = −0.037, *p* = 0.728; no significant association between HbA1c and PI, *r* (88) = −0.033, *p* = 0.757; HbA1c and PD, *r*(88) = 0.045, *p* = 0.677; HbA1c and CAL, *r* (88) = −0.057, *p* = 0.59 was revealed. There was a significant positive association between GI and PI, *r* (88) = 0.943, *p* = 0. There was no significant association between GI and PD, *r* (88) = 0.095, *p* = 0.372; GI and CAL *r* (88) = −0.037, *p* = 0.727; PI and PD, *r* (88) = 0.101, *p* = 0.342; PI and CAL, *r* (88) = −0.008, *p* = 0.942, and between PD and CAL, *r* (88) = −0.009, *p* = 0.932. At 6 months, the results of the Spearman rang correlation indicated that there was no significant association between HbA1c and periodontal indices. No statistically significant correlation between CRP and all the variables was found.

A statistically significant positive association was found only between PI and GI, *r* (88) = 0.76, *p* = 0, and GI and CAL, *r* (88) = 0.235, *p* = 0.025 (Table 5).

### 3.2. Comparison of Each Parameter between Groups at Baseline, 3 and 6 Months

As shown in Table 6, at baseline, the intervention group had higher values of HbA1c (*Mdn* = 7.7) as the control group (*Mdn* = 7.5).

Mann–Whitney *U*-test showed this difference was not statistically significant, *U* = 3932, *p* = 0.735, *r* = 0.025. *p*-value equals 0.736524, (*p* (x ≤ Z) = 0.368262). This means that if we would reject H0, the chance of type I error (rejecting a correct H0) would be too high: 0.7365 (73.65%). The larger the *p*-value the more it supported H0. The statistic Z test, at baseline, was −0.336460 and was in the 95% critical value accepted range: [−1.9600: 1.9600]. U = 3932.00, was in the 95% accepted range: [3365.5300: 0.01067]. The statistic S’ equals 349.224. The observed standardized effect size, Z/√(*n*1+*n*2), was small: 0.025, which indicates that the magnitude of the difference between the probability to choose a bigger value from the intervention group and the probability to choose a bigger value from the control group was small. The common language effect size, *U*1/(*n*1*n*2), was 0.49. This was the probability that a random value from the intervention group was greater than a random value from the control group. At 3 months, the intervention group displayed lower values of HbA1c (*Mdn* = 7.25) than the control group (*Mdn* = 7.65), but the difference was not statistically significant, *U* = 3512, *p* = 0.123, *r* = 0.115. *p*-value equals 0.123723, (*p* (x ≤ Z) = 0.0618613). The test statistic Z equals −1.539335; *U* = 3512.00. The statistic S’ was 349.177, the statistic Z test was −1.539335 and was in the 95% critical value accepted range: [−1.9600: 1.9600]. *U* = 3512.00 and was in the 95% accepted range: [3365.6300: 0.01067]. The observed standardized effect size, Z/√(*n*1+*n*2), was small (0.11). The common language effect size, *U*1/(*n*1*n*2), was 0.43. At 6 months, the intervention group had lower values of HbA1c (*Mdn* = 7.2) than the CG (*Mdn* = 7.3). The difference was not statistically significant, *U* = 3757, *p* = 0.401, *r* = 0.063. *p*-value equals 0.474786, (*p* (x ≤ Z) = 0.762607). The test statistic Z equals 0.714714; *U* = 4253.00. The statistic S’ equals 346.292. The observed standardized effect size, Z/√(*n*1+*n*2), was small (0.053). The common language effect size, *U*1/(*n*1*n*2), was 0.53.

The PI at baseline for the intervention group had higher values (*Mdn* = 80.5) than the control group (*Mdn* = 70.5). These results showed this difference big enough to be statistically significant, *U* = 2642.5, *p* = 0, *r* = 0.301. *p*-value equaled 0.0000527901, (*p* (x ≤ Z) = 0.999974) and the chance of type1 error was small: 0.00005279 (0.0053%). The statistic Z test equaled 4.042918 was not in the 95% critical value accepted range: [−1.9600: 1.9600]. *U* = 2642.5 was not in the 95% accepted range: [3367.9000: 0.01067]. The statistic S’ equals 348.016. The observed standardized effect size, Z/√(*n*1+*n*2), was medium (0.30). The common language effect size, U1/(*n*1*n*2), was 0.67. At three months, the intervention group had lower values of PI (*Mdn* = 15.5) compared to the control group (*Mdn* = 70), and the difference was statistically significant, *U* = 57.5, *p* = 0, *r* = 0.852. *p*-value equals 0.00000, (*p* (x ≤ Z) = 0.00000). The test statistic Z equaled −11.427513; *U* = 57.50. The statistic S’ equaled 349.332. At six months of follow up the difference of PI between the groups (*Mdn* = 12 for IG and *Mdn* = 69 for CG respectively) was statistically significant, *U* = 15, *p* = 0, *r* = 0.861. *p*-value equals 0.00000, (*p* (x ≤ Z) = 0.00000). The test statistic Z equaled −11.548199; *U* = 15.00. The statistic S’ equaled 349.362. The observed standardized effect size, Z/√(*n*1+*n*2), was large (0.86). The common language effect size, *U*1/(*n*1*n*2), was 0.0019. At baseline, also the difference of GI between the groups was statistically significant, *U* = 3330.5, *p* = 0.039, *r* = 0.154. Specifically, the IG had higher values (*Mdn* = 67) than the control group (*Mdn* = 65). *p*-value equaled 0.0377597, (*p* (x ≤ Z) = 0.981120). This means that the chance of type1 error was small: 0.03776 (3.78%). The test statistic Z equaled 2.077453 and was not in the 95% critical value accepted range: [−1.9600: 1.9600]. *U* = 4776.00 was not in the 95% accepted range: [3365.5300: 0.01067]. The statistic S’ equaled 349.226. The observed standardized effect size, Z/√(*n*1+*n*2), was small (0.15). The common language effect size, *U*1/(*n*1*n*2), was 0.59. At 3 months, the intervention group showed lower values of GI (*Mdn* = 12.5) in comparison with the control group (*Mdn* = 65). The difference was statistically significant, *U* = 70, *p* = 0, r = 0.849. *p*-value equaled 0.00000, (*p* (x ≤ Z) = 0.00000). The test statistic Z equaled −11.38; *U* = 70. The statistic S’ equaled 349.392. The observed standardized effect size, Z/√(*n*1+*n*2), was large (0.85). The common language effect size, *U*1/(*n*1*n*2), was 0.0086. At 6 months, data showed that the GI values were lower in the intervention group (*Mdn* = 10) than the control group (*Mdn* = 64.5). Mann–Whitney *U*-test showed this difference was statistically significant, *U* = 33, *p* = 0, *r* = 0.857. *p*-value equaled 0.00000, (*p*(x ≤ Z) = 0.00000). This means that the chance of type1 error was small: 0.000 (0.0%). The smaller the *p*-value the more it supported H1. The test statistic Z equaled −11.49; U = 33. The statistic S’ equaled 349.322. The observed standardized effect size, Z/√(*n*1+*n*2), was large (0.86). The common language effect size, *U*1/(*n*1*n*2), was 0.0041. The PD value at baseline was higher for IG (*Mdn* = 4.85) than the CG (*Mdn* = 4.5). This difference was statistically significant, *U* = 2885, *p* = 0.001, *r* = 0.249. At 3 months, IG group revealed lower values (*Mdn* = 2.9) than the control group (*Mdn* = 4.5). The difference was statistically significant, *U* = 42.5, *p* = 0, *r* = 0.855. Finally, at six months, the IG exhibited decreased values of PD (*Mdn* = 2.86) than the CG group (*Mdn* = 4.605), showing a statistically significant difference, *U* = 2, *p* = 0, *r* = 0.864. The CAL parameter, at baseline, was higher for IG (*Mdn* = 5.3) than the CG (Mdn = 4.8). This difference indicated a statistically significant difference, *U* = 2625, *p* = 0, *r* = 0.305. At the second timeline (3 months), the IG group presented lower values (*Mdn* = 4.1) than the control group (*Mdn* = 4.8). This difference was statistically significant, *U* = 1307.5, *p* = 0, *r* = 0.585. The CAL index at 6 months had lower values (*Mdn* = 4) than the CAL for CG (*Mdn* = 5), establishing a statistically significant differences between groups, *U* = 948, *p* = 0, *r* = 0.662. The CRP at baseline had higher concentration (*Mdn* = 2.3) for the intervention group compared to the control group (*Mdn* = 2.145), but the magnitude of the difference was not statistically significant, *U* = 4480.00, *p* = 0.222, *r* = 0.091. The test statistic Z equaled 1.2; *U* = 44.8. The statistic S’ equaled 349.437. The observed standardized effect size, Z/√(n1+n2), was small (0.092). The common language effect size, *U*1/(*n*1*n*2), was 0.55. At 3 months, the difference between the groups was big enough to be statistically significant. *p*-value equaled 0.0000467009, (*p* (x ≤ Z) = 0.0000233504). This means that the chance of a type1 error was small: 0.00004670 (0.0047%). The test statistic Z equaled −4.071552 and was not in the 95% critical value accepted range: [−1.9600: 1.9600]. *U* = 2627.00 and was not in the 95% accepted range: [3365.2400: 0.01067]. The statistic S’ equaled 349.375. At 6 months, the control group had higher values (*Mdn* = 2.33) than the intervention group (*Mdn* = 1.32). This difference was statistically significant, *U* = 1839.5, *p* = 0, *r* = 0.472. *p*-value equaled 2.52750 × 10^−10^, (*p* (x ≤ Z) = 1.26375 × 10^−10^). This means that the chance of type1 error (rejecting a correct H_0_) was small: 2.527 × 10^−10^ (2.5 × 10^−8^%). The test statistic Z equaled −6.325; *U*=18.39. The statistic S’ equaled 349.391. The observed standardized effect size, Z/√(*n*1+*n*2), was medium (0.47). The common language effect size, *U*1/(*n*1*n*2), was 0.23.

### 3.3. Difference of HbA1c Values Within the Intervention Group Over the 6 Months

At baseline, the HbA1c values of the intervention group were higher (*Mdn* = 7.7) than the HbA1c at 3 months (*Mdn* = 7.25). Mann–Whitney *U*-test showed this difference was not statistically significant, *U* = 3429, *p* = 0.075, *r* = 0.133, while the difference between HbA1c values at baseline (*Mdn* = 7.7) and 6 months (*Mdn* = 7.2) was statistically significant, *U* = 3174.5, *p* = 0.012, *r* = 0.187. No statistically significant difference was revealed about the differences between the HbA1c values at 3 months (*Mdn* = 7.25) and the HbA1c values at 6 months (*Mdn* = 7.2), *U* = 3844.5, *p* = 0.556, *r* = 0.044.

### 3.4. Difference of HbA1c Values Within the Control Group Over the 6 Months

At baseline, the HbA1c values of the control group were lower (*Mdn* = 7.7) compared with the analysis at 3 months (*Mdn* = 7.9), but this difference was not statistically significant, *U* = 3751.5, *p* = 0.393, *r* = 0.064. At 6 months, the control group had lower values (*Mdn* = 7.3) than the baseline (*Mdn* = 7.7), showing a statistically significant difference *U* = 3149.5, *p* = 0.01, *r* = 0.192. Finally, at 6 months, the control group had lower values (*Mdn* = 7.3) than the 3 months evaluation (*Mdn* = 7.3), showing a statistically significant difference, *U* = 3142, *p* =0.009, *r* = 0.194.

### 3.5. Difference of CRP Values Within the Intervention Group Over the 6 Months

At 3 months the IG registered lower values (*Mdn* = 1.43) than the baseline (*Mdn* = 2.12). This difference was statistically significant, *U* = 3209.5, *p* = 0.016, *r* = 0.179. At 6 months, results show lower values (*Mdn* = 1.32) than the baseline (*Mdn* = 2.12). The difference was statistically significant, *U* = 2606.5, *p* = 0, *r* = 0.308. The difference between 3 and 6 months was not statistically significant, *U* = 3460.5, *p* = 0.092, *r* = 0.126, with *Mdn* = 1.43 at 3 months and *Mdn* = 1.32 at 6 months.

### 3.6. Difference of CRP Values Within the Control Group Over the 6 Months

At 3 months, the control group had higher values (*Mdn* = 2.5) than the baseline (*Mdn* = 2.12), but the difference was not statistically significant, *U* = 3492.5, *p* = 0.111, *r* = 0.119. At 6 months, the CRP concentration had higher values than the baseline (*Mdn* = 2.12), showing an *Mdn* of 2.33. This difference was not statistically significant, *U* = 3806, *p* = 0.487, *r* = 0.052.

The difference between the value at 3 months and 6 months was not statistically significant, *U* = 3610.5, *p* = 0.21, *r* = 0.094, with *Mdn* = 2.33 and *Mdn* = 2.5, respectively.

### 3.7. Effect on HbA1c

A multiple linear regression analysis was performed at 3 and 6 months to examine whether the periodontal parameters significantly predicted HbA1c in the intervention group. At 3 months analysis, the regression model indicated that the predictors explained 0.056 of the variance and a collective significant effect was not found. F = 014, *p* = 0.96, R^2^ = 0.0068. Since *p*-value ≥ α (0.5), we accepted the H_1_. The adjusted R square equaled −0.0399332. The coefficient of multiple correlations (R) equaled 0.0824947. It means that there was a very weak direct relationship between the predicted data (ŷ) and the observed data (y). The individual predictors are shown in Table 7.

### 3.8. Effect on CRP

A multiple linear regression analysis was performed to examine whether the, periodontal indices and glycated hemoglobin variables significantly predicted CRP over 3 and 6 months. The regression model indicated that the predictors explained 0.03 of the variance and a collective significant effect was not found. F = 0.522, *p* = 0.759, R^2^ = 0.03. The individual predictors result is shown in Table 8.

## 4. Discussion

Diabetes is a complex metabolic disorder affecting the glucose status of the human body, characterized by impaired action, secretion of insulin or both, resulting in hyperglycemia [12,13,17,18,19,21,22,34]. Chronic hyperglycemia related to diabetes is associated with end organ failure. Majority of people with diabetes fall into two broad pathogenetic categories, type 1 or type 2 diabetes [23,35]. The treatment in chronic illness has a decisive role and function of the economy in improving the quality of living [24,25,26,27,35,36,37]. The treatment of a patient with diabetes requires consideration of key pathogenic characteristics, the duration of disease, the age of patient and the presence of secondary diagnoses for specific complications or comorbidities [38]. Priority in management diabetes is the regression of pathology, by reaching the euglycemia and controlling the metabolic alterations [5,39,40,41,42]. The role of inflammation is under intensive scrutiny with several clinical trials to have been completed while more are in development [43]. The effective management of diabetes by introduction of the insulin therapy has been able to modify the natural history of diabetes, but there is currently no evidence that they prevent the long-term sequelae [44,45,46]. Conversely, the prolonged disease has a detrimental effect on the organic system, leading to significant chronic complications [47]. As a consequence of the glycemic excursions, diabetes has the major complications in the vascular system [48,49,50]. The micro- (retinopathy, nephropathy and neuropathy and periodontal disease) and macrovascular (coronary, peripheral and cerebral vasculopaty), progressive injury occurs in at least 76% of diabetic patients within 10 years follow up [13,44,45,46]. Previously studies reported the observation of characteristic microvascular changes in the periodontal tissues associated with diabetic complications [39,51,52,53,54,55].

Periodontitis is chronic infectious inflammatory disease characterized by the progressive destruction of the periodontal apparatus [25,26,27,28] depending on the complex relationship between a susceptible host, pathogenic bacteria and an environment propitious for disease progression. Clinical presentation will be the same as for any extent, varying from minor to highest gum’s bleeding, swelling and pain. Dental plaque, an organized biofilm of microorganisms, is the essential noxa pathogena, which elicits the onset of gingival inflammation [29,30], establishing and maintaining the intimate host–pathogen association. It begins as low-grade, protracted response to oral pathogen colonization, characterized by prolonged activation of a large amount of mononuclear leukocytes (monocytes and lymphocytes) accompanying tissue damage due to the vicious cycle linking inflammation and the pathological process it accompanies. When the pathogen stimulus is not controlled it can trigger a systemic response that has been demonstrated to prolong the general inflammatory status. Evidence has consistently supported that induction of cytokines, chemokines and acute-phase reactants occur during periodontal infection and is completely dependent on disease persistence [31]. Increased levels of C-reactive protein (CRP) have been associated with periodontitis in systemically healthy subjects.

It is well known the biunivocal association between periodontal disease and diabetes and pathophysiological mechanism related to immune functioning but is an under-recognized clinical problem [39,52,53,54,55,56,57]. Non-surgical periodontal treatment is associated with the reduction of 0.4% glycate hemoglobin (HbA1c) approximately [10,58,59,60]. The concept of periodontitis as “the sixth complication of diabetes” has been popularized by Loe in 1993 [49], who firstly demonstrated that the prevalence of periodontal disease was three times higher among type II diabetic persons compared with the nondiabetic [39,52,53]. Severe periodontitis has been associated with poor controlled diabetic condition, confirming that the release of inflammatory mediators due to poor glycemic control is implicated in the pathogenesis of periodontitis [5,23]. The experimental association studies have increased our knowledge about the link about diabetes and periodontitis. Non-surgical periodontal treatment has been found to be beneficial in glycemic control depending on several factors [26,49]. Consistent with findings in the literature, our study examined the correlation between periodontitis and later improvement of glycated hemoglobin percentage after non-surgical periodontal treatment, and the reduction of systemic inflammation through the assessment of C-reactive protein concentration [32,33,34,35]. Since previous studies clearly showed that periodontal therapy is effective in reducing the HbA1c level [16,20,32,33], we hypothesized that periodontal status may adversely affect diabetes. Thus, we suggest that periodontal treatment may improve metabolic control in type 2 diabetes. In terms of study limitations, long-term follow-up might have offered greater awareness. Natural history studies indicate that alteration of the HbA1c level is determined by several factors. The hypothesis that periodontal treatment has a role in influencing rates HbA1c level remains controversial. The results from this study did not completely clarify the potential role of periodontal therapy in the metabolic control of diabetic patients. The high proportion of patients who demonstrated HbA1c reduction over 6-months 2 years in both groups indicates that several factors contribute to fluctuations of glycated hemoglobin. Our study found no statistical differences between the periodontal treatment or not in the metabolic control of diabetic patients over 6 months. Both groups achieved similar fluctuations over 6 months. Specifically, periodontal indices were elevated both in the intervention and control group at baseline, but a strong correlation was not found with HbA1c level. The intervention group initially had a higher level of HbA1c and, therefore, the treatment effect was more significant. It is a good result for periodontal treatment. About the CRP concentration, we aimed to assess the oscillation of concentration after the periodontal treatment, assuming that the periodontal status could be a predictor of its decrease.

It is generally accepted that inflammation cytokines like the C-reactive protein (CRP) are a determinant for the connection between diabetes and periodontitis [61]. A meta-analysis conducted by Teeuw et al. [62] showed a significant reduction of CRP level after the periodontal treatment, and Katagiri et al. [63] observed a strong relationship between the change of CRP and HbA1c concentration. Several large scale cross-sectional studies reported elevated levels of serum CRP in gingivitis and periodontitis [64,65,66]. The effectiveness of periodontal treatment on CRP levels was also investigated from several reports indicating that CRP was consistently elevated in periodontitis individuals (>2.1 mg/L) compared with healthy controls. Our results were consistent with previous funding, which reported that periodontal treatment determined a statistically significant decrease of the CRP plasma levels after 3 months. It has been documented that 3 months post treatment is a suitable interval for the primary evaluation of non-surgical periodontal treatment. Therefore, our data confirm that periodontal therapy could significantly reduce systemic inflammation by improvement of the periodontal status.

However, also our results show that the CRP value was plausibly elevated in patients with periodontitis and diabetes, but no statistically significant relationship between all the variables was found. While more is becoming known regarding the role of inflammatory pathways linking diabetes and periodontitis, there remain many open questions [11,12,13,17,18,19,21,22,54].

## 5. Conclusions

There is emerging evidence to support the existence of a two-way relationship between diabetes and periodontitis, with diabetes increasing the risk for periodontitis, and periodontal inflammation negatively affecting glycemic control [39,59]. The mechanisms that underpin the links between these two conditions are not completely understood, but involve aspects of immune functioning, neutrophil activity and cytokine biology. One of the main findings of our study was that poor glycemic control was related to severity of periodontitis [61] and that periodontal disease could increase the amount of plasma CRP [10,67,68,69]. Further studies are needed for insight into the molecular mechanisms underlying inflammation and to elaborate further on the physiological role of this phenomenon.

## Figures and Tables

**Figure 1 ijerph-18-03018-f001:**
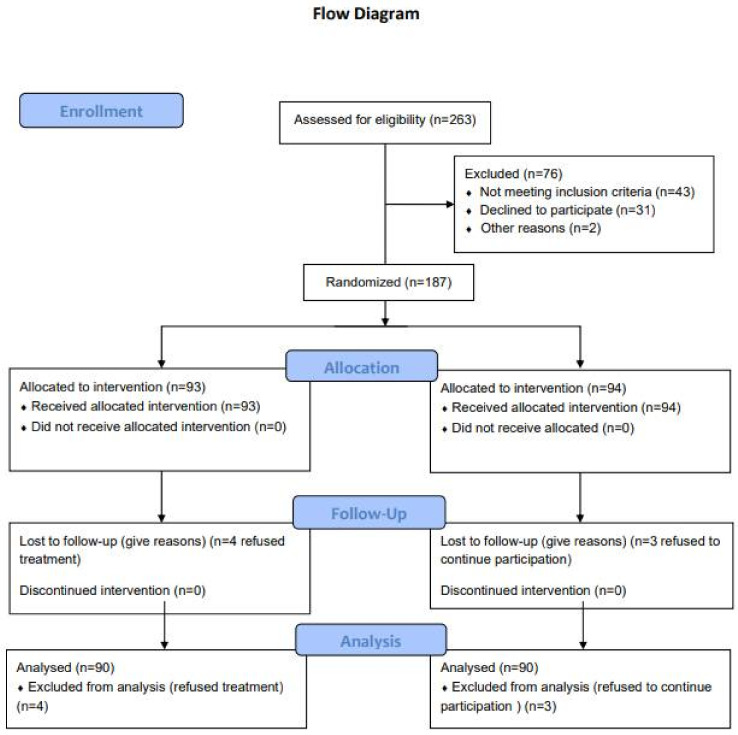
CONSORT flow chart. This figure shows the flow of patients through the trial according to the criteria recommended in the CONSORT guidelines.

**Table 1 ijerph-18-03018-t001:** Participants’ characteristics.

Characteristics	Intervention Group	Control Group
**Sex (*N*)**		
**Female** **Male**	50 40	54 36
**Age (y) mean ± SD**	53.2 ± 11.2	56. ± 6.9
**Average *BMI (kg/m^2^, mean ± SD)**	27.8 ± 6.3	22.3 ± 5.1

* BMI: Body Mass Index.

**Table 2 ijerph-18-03018-t002:** Descriptive analysis of clinical parameters at baseline of both groups.

	HbA1_c_ (%) *IG	HbA1_c_ (%) **CG	*p* Value	CRP mg/L IG	CRP mg/L CG	*p* Value	PI (mm) IG	PI (mm) CG	*p* Value	GI (%) IG	GI (%) CG	*p* Value	PD (mm) IG	PD (mm) CG	*p* Value	CAL (mm) CG	CAL (mm) IG	*p* Value
**Mean**	8.081	8.767	0.048	2.514	2.302	0.431	80.8	69.067	0.011	68.744	62.244	0.74	4.851	4.572	0.037	4.883	5.178	0.039
**Median**	7.7	7.5	-	2.3	2.145	-	80.5	70.5	-	67	65	-	4.85	4.5	-	4.8	5.3	-
**Std. Deviation**	1.965	8.514	-	1.173	1.221	-	16.348	19.886	-	18.757	21.439	-	0.568	0.482	-	0.513	0.339	-
**Variance**	3.861	72.484	-	1.375	1.492	-	267.26	395.456	-	351.83	459.647	-	0.323	0.233	-	0.263	0.115	-
**Minimum**	5.5	5.5	-	0.26	0.26	-	45	21	-	29	18	-	4	3.3	-	3.82	4.5	-
**Maximum**	17	88	-	5.43	5.11	-	100	100	-	100	100	-	8	5.8	-	6.5	5.9	-
**IQR**	1.375	1.525	-	1.405	1.91	-	33	32.2		29.5	31..2	-	0.7	0.8	-	0.5	0.7	-
**Skew**	2.592	9.258	-	0.215	0.513	-	−0.314	−0.2	-	−0.046	0.108	-	1.831	0.269	-	0.107	−0.162	-
**Kurtosis**	8.741	87.062	-	−0.213	−0.702	-	−1.166	−0.712	-	−0.946	−0.783	-	9.267	−0.392	-	0.082	−0.784	-

*IG: Intervention Group; **CG: Control Group; Hb: Hemoglobin; CRP: C-reactive protein; PD: Pocket Depth; CAL: Clinical Attachment Level; BOP: Bleeding on Probing; GI: Gingival Index; PI: Plaque Index; IQR: Interquartile range *p* < 0.05.

**Table 3 ijerph-18-03018-t003:** Correlation and significance analysis between all the variables of the intervention group at baseline.

Baseline		HbA1c	PI	GI	PD	CAL	CRP
**HbA1c**	Spearman correlation	1	0.106	0.126	−0.136	−0.142	−0.222
	*p*-value (2-tailed)		0.322	0.237	0.201	0.181	0.093
**PI**	Spearman correlation	0.106	0.09	0.123	0.091	0.938	0.224
	*p*-value (2-tailed)	0.322		0	0.094	0.232	0.091
**GI**	Spearman correlation	0.126	0.914	1	0.12	0.112	−0.205
	*p*-value (2-tailed)	0.237	0		0.259	0.293	0.123
**PD**	Spearman correlation	−0.136	0.177	0.12	1	0.718	−0.224
	*p*-value (2-tailed)	0.201	0.094	0.259		0	0.123
**CAL**	Spearman correlation	−0.142	0.127	0.112	0.718	1	−0.01
	*p*-value (2-tailed)	0.181	0.232	0.293	0		0.938
**CRP**	Spearman correlation	−0.222	−0.224	−0.205	−0.224	−0.01	1
	*p*-value (2-tailed)	0.093	0.091	0.123	0.091	0.938	0

**Table 4 ijerph-18-03018-t004:** Correlation and significance analysis between all variables of the intervention group at 3 months.

3 Months		HbA1c	GI	PI	PD	CAL	CRP
**HbA1c**	Spearman correlation	1	−0.037	−0.033	0.045	−0.057	−0.027
	p-value (2-tailed)		0.728	0.757	0.677	0.59	0.799
**GI**	Spearman correlation	−0.037	1	0.943	0.095	−0.037	0.131
	p-value (2-tailed)	0.728		0	0.372	0.727	0.217
**PI**	Spearman correlation	−0.033	0.943	1	0.101	−0.008	0.127
	p-value (2-tailed)	0.757	0		0.342	0.942	0.233
**PD**	Spearman correlation	0.045	0.095	0.101	1	−0.009	−0.067
	p-value (2-tailed)	0.677	0.372	0.342		0.932	0.53
**CAL**	Spearman correlation	−0.057	−0.037	−0.008	−0.009	1	0.103
	p-value (2-tailed)	0.59	0.727	0.942	0.932		0.335
**CRP**	Spearman correlation	−0.027	0.131	0.127	−0.067	0.103	1
	p-value (2-tailed)	0.799	0.217	0.233	0.53	0.335	

**Table 5 ijerph-18-03018-t005:** Correlation and significance analysis between the HbA1c level and periodontal parameters of the intervention group at 6 months.

6 months		HbA1c	PI	GI	PD	CAL	CRP
**HbA1c**	Spearman correlation	1	−0.032	0.058	−0.03	−0.034	0.046
	*p*-value (2-tailed)		0.767	0.587	0.779	0.753	0.669
**PI**	Spearman correlation	−0.032	1	0.76	0.095	0.168	0.11
	*p*-value (2-tailed)	0.767		0	0.375	0.113	0.301
**GI**	Spearman correlation	0.058	0.76	1	−0.053	0.235	0.118
	*p*-value (2-tailed)	0.587	0		0.62	0.025	0.268
**PD**	Spearman correlation	−0.03	0.095	−0.053	−0.152	−0.152	−0.05
	*p*-value (2-tailed)	0.779	0.375	0.62	−0.152	0.153	0.639
**CAL**	Spearman correlation	−0.034	0.168	0.235	−0.152	1	0.091
	*p*-value (2-tailed)	0.753	0.113	0.025	0.153		0.396
**CRP**	Spearman correlation	0.046	0.11	0.118	−0.05	0.091	1
	*p*-value (2-tailed)	0.669	0.301	0.268	0.639	0.396	

**Table 6 ijerph-18-03018-t006:** Comparison between groups at each time point.

Indices	Mann-Whitney U	W of Wilcoxon	Z	Asymptotic Significance (2-tailed)	Exact Significance (2-tailed)
**HbA1c**					
Baseline	3.93	1.56	−0.33	0.73	0.73
3 months	3.08	2.26	−2.77	0.006	0.006
6 months	3.75	1.49	−0.83	0.4	0.4
**PI**					
Baseline	2.64	2.46	−4.04	0	0
3 months	57.5	3.00	−11.4	0	0
6 months	15	0	−11.5	0	0
**GI**					
Baseline	3.33	2.44	−2.06	0.039	0.04
3 months	70	12.0	−11.39	0	0
6 months	33	1.00	11.49	0	0
**PD**					
Baseline	2.88	2.15	−3.3	0.001	0.001
3 months	42.5	0	−11.4	0	0
6 months	2	0	−11.5	0	0
**CAL**					
Baseline	2.62	2.32	−4.08	0	0
3 months	1.30	6.25	−7.85	0	0
6 months	9.48	2.84	−8.8	0	0
**CRP**					
Baseline	3.62	4.48	1.22	0.222	0.221
3 months	2.62	2.12	−4.07	0	0
6 months	1.83	1.83	−6.32	0	0

**Table 7 ijerph-18-03018-t007:** The individual predictors.

3 Months	Unstandardized Coefficients	Standardized Coefficients			
Model	B	β	Standard Error	t	*p*-Value
(Constant)	13.9		3.23	4.32	0
PI	−0.03	−0.32	0.02	−1.3	0.18
GI	0.03	0.32	0.02	1.36	0.17
PD	0.27	0.08	0.45	0.61	0.5
CAL	−1.24	−0.21	0.74	−1.66	0.1

**Table 8 ijerph-18-03018-t008:** The individual predictors.

	Unstandardized Coefficients	Standardized Coefficients			
Model	B	Beta	Standard error	t	*p*-Value
(Constant)	1.37		1.01	1.348	0.181
HbA1c	0.01	0.01	0.07	0.16	0.873
PI	−0.006	−0.05	0.05	−0.125	0.901
GI	0.02	0.21	0.05	0.455	0.651
PD	−0.08	−0.05	0.1	−0.468	0.641
CAL	0.11	0.07	0.16	0.67	0.504

## Data Availability

All data to support the findings of this study are available contacting the corresponding author upon request. The authors have annotated the entire data building process and empirical techniques described in the paper.

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
