# Peer review of "Inflammatory Status and Glycemic Control Level of Patients with Type 2 Diabetes and Periodontitis: A Randomized Clinical Trial"

_ijerph, 2021, doi:10.3390/ijerph18063018_

Round 1

Reviewer 1 Report

Dear Authors,

  1. The study is clear, very well conducted and very relevant in the clinical setting. The title is relevant and the references are recent.
  2. The research question is clearly outlined and justifies
  3. The study methods are valid and reliable. The good conduct of the study can allow its replication.
  4. The results are appropriate. The data is presented in appropriate way (tables and figures are relevant and clearly presented. Titles and columns are correctly and clearly).I’m not unable to provide further statistical opinions.
  5. The results are widely discussed and the discussion of the study has been extensively covered.

The topic dealt with is fundamental for clinicians to allow a better diagnosis and treatment of the patient, for this reason further studies will be necessary to deepen the molecular mechanisms of inflammation.

Author Response

Reviewer 1:

Dear Authors,

  1. The study is clear, very well conducted and very relevant in the clinical setting. The title is relevant and the references are recent.
  2. The research question is clearly outlined and justifies
  3. The study methods are valid and reliable. The good conduct of the study can allow its replication.
  4. The results are appropriate. The data is presented in appropriate way (tables and figures are relevant and clearly presented. Titles and columns are correctly and clearly). I’m not unable to provide further statistical opinions.
  5. The results are widely discussed and the discussion of the study has been extensively covered.

The topic dealt with is fundamental for clinicians to allow a better diagnosis and treatment of the patient, for this reason further studies will be necessary to deepen the molecular mechanisms of inflammation.

Authors: Thank you so much. We are flattered!

Reviewer 2 Report

The aim of this manuscipt is to recover the effect of non-surgical therapy on the glycaemic control level of T2DM patients with periodontitis. The design of this study is appropriate and it is interesting for the our clinical work.  There are some conments as follows:

  1. The introduction of this manuscript is tedious. In introduction, you should only introduce type 2 Diabetes and periodontitis and the relationship between these two diseases, show the clinical trials about the relationship between diabetes and periodontitis and finally explain why you do this study and its significance.
  2. The results showed that HbA1c, PI ,PD and CAL were significantly different between groups at baseline, how do you adjust these values when you compared the values in different time points.
  3. Discuss the results in detail and point out the shortcomings of this study in discussion part.

Author Response

Reviewer 2:

Reviewer 2:

The aim of this manuscipt is to recover the effect of non-surgical therapy on the glycaemic control level of T2DM patients with periodontitis. The design of this study is appropriate and it is interesting for the our clinical work.  There are some conments as follows:

  1. The introduction of this manuscript is tedious. In introduction, you should only introduce type 2 Diabetes and periodontitis and the relationship between these two diseases, show the clinical trials about the relationship between diabetes and periodontitis and finally explain why you do this study and its significance.

Authors: The introduction has been modified, as requested. The new text is the following:

"Despite advances in prevention and treatment measures, diabetes mellitus remains one of the most important chronic diseases worldwide, associated with high mortality and morbidity, with an estimated 300 million cases in the year 2025 [1]. Research has pointed to the multiplicative effect when several risk factors are present and, of interest, scientific evidence underlines the possibility of closely linked inflammatory etiology. Accumulating evidence supports a key role of periodontal inflammation in the pathophysiology of several systemic disorders [2-4]. Progress in understanding the pathogenesis of diabetes has generated increasing interest in targeting inflammatory pathways and biomarkers of inflammation to help prevent and control diabetes and related sequelae [5-8]. Periodontitis is a chronic multifactorial inflammatory disease affecting the tissues surrounding the teeth [7], and has a prevalence estimated at 20% to 50% of the global population. Commonly, over 10% of older population have severe disease [8-10]. Periodontitis can promote systemic chronic inflammation (SCI) that can, in turn, lead to exacerbation of type 2 diabetes [9-11]. This consistent clinical evidence for an interconnection between periodontitis and diabetes risk comes from multiple randomized controlled trials (RCTs) and epidemiological investigations data that have proven that periodontal treatment resulted in HbA1c levels reduction [10,12,13]. All these findings are in harmony with the hypothesis that periodontitis and diabetes may fuel each other in a bidirectional pathway [9,13], in which a reciprocal action of inflammatory mediators leads to worsen of pathological conditions, becoming closely linked diseases [11].

The metabolic dysregulation in diabetes associated with hyperglycaemia and hyperinsulinemia are predisposing factors to infectious diseases in diabetes. Periodontitis is one of the most common complications of diabetes mellitus, affecting more of 50% of the diabetic population [28]. Periodontitis has been implicated in an increased risk of type 2 diabetes, and diabetes patients are two-five times as likely to develop risk of suffering periodontitis compared to healthy individuals, but the molecular mechanisms linking these two diseases are partially known [29,30,5] Mechanistic studies in in vivo conditions agrees to assert that among regulatory factors of this connection should be explored [31]. Specifically, pathway model that aggregates the knowledge about the impact of diabetes on periodontal status have been well-documented and characterized [28]. Data on the clinical impact of diabetes on periodontal inflammation enhancement come from two sources: the potential direct effect of hyperglycaemic status which may be responsible for the oral microbiome shifting, impaired cellular function and host defence, and upregulation of the circulant proinflammatory mediators known to be released primarily by immune cells [18,23]; the accelerated glycation of protein and lipids mediated by AGES and fatty acids which may indirectly contribute to degenerative process of periodontal tissues and influence negatively osteogenesis [30-33]. In contrast, it has been assumed that long term low-level exposure to periodontal inflammation have a negative impact on metabolic control, but the pathway by which periodontitis contributes to worsening diabetes is still unclear [30]. Human studies have demonstrated that patients suffering from periodontitis exhibit higher concentrations of pro-inflammatory cytokines- such as tumor necrosis factor-alpha (TNF-α), C-reactive protein (CRP), interleukin-6 (IL-6), Il-1β. Several mechanisms of altered macrophage polarization during obesity have been recently suggested. Based on this evidence, dysregulation in the cytokine network may contribute to development or sustaining diabetes via activation of the chronic systemic inflammation [17,28-30]. The clinical relevance of increases of CRP in subjects with periodontitis has been demonstrated, by showing that the peripheral inflammatory response is reflected in increased concentrations of C-reactive Protein in patients with periodontal infections compared to periodontally health individuals. The most reliable evidence for the co-occurrence of diabetes and periodontitis and the correlation between the two diseases is a function of inflammatory pathway [2,5,13,26].

Based on the simultaneous occurrence of periodontitis and diabetes, as recorded in most of clinical trials, this paper briefly resumes the conceptual cycle that accounts for the theory of bidirectional relationship between diabetes and periodontitis and explores the impact of suppression of periodontal inflammatory response pathway in minimising the interrelation between the peripheral and systemic inflammatory status. In this perspective, the chief object of this trial was aimed to investigate the effectiveness of non-surgical periodontal treatment on glycaemic control in patients diagnosed type 2 diabetes and, simulyaneously, the potential improvement of systemic inflammatory status."

  1. The results showed that HbA1c, PI, PD and CAL were significantly different between groups at baseline, how do you adjust these values when you compared the values in different time points.

Authors: The difference at baseline were just the starting point. The our interest was to evaluate the fluctuation of each measurement/parameter after the non surgical periodontal treatment. The magnitude of the difference between groups had a significance in terms of the effect of treatment.

  1. Discuss the results in detail and point out the shortcomings of this study in discussion part.

Authors: We are sorry, but we have largely described the results, by dividing for each parameter. We have, as requested, added a brief test about the results in discussion part, as follows: "It is generally accepted that inflammation cytokines like C-reactive protein (CRP) are determinant for the connection between diabetes and periodontitis. A meta-analysis conducted by Teeuw et al.  showed a significant reduction of CRP level after periodontal treatment, as well as Katagiri et al.  observed a strong relationship between the change of CRP and HbA1c concentration. Several large scale cross-sectional studies reported elevated levels of serum CRP in gingivitis and periodontitis. The effectiveness of periodontal treatment on CRP levels was also investigated from several reports indicating that CRP is consistently elevated in periodontitis individuals (>2.1 mg/l) compared with healthy controls. Our results are consistent with previous funding which reported that periodontal treatment determined a statistically significant decrease of the CRP plasma levels after 3 months. It has been documented that 3 months post treatment is a suitable interval for the primary evaluation of non-surgical periodontal treatment. Therefore, our data confirm that periodontal therapy can significantly reduce systemic inflammation by improvement of the periodontal status."

Reviewer 3 Report

Dear authors,  

This is an interesting article but some aspects need to be explained.

Broad and comprehensive introduction, although a little long. Some aspects should be pointed out:

material and methods

  • The classification of periodontal disease must be updated. According to the description, the classification of 2018 was used, not 1999.
  •  It is not mentioned if the person who performed the periodontal examinations was a trained and calibrated examiner.
  • the statement that patients with insulin-dependent diabetes were excluded is mistaken, since patients with DM2 may also need insulin therapy. It must be said that patients with DM1 or other types were excluded
  • It is mentioned that smokers were excluded. what about ex-smokers?
  • What is the dosage method for HbA1c? Is it certified?
  • What is the CRP dosage method?
  • it is not clear how the median for plaque and gingival indexes was obtained. Why was a dichotomous index not used, such as the presence or absence of biofilm and the presence or absence of marginal bleeding? the use of an index that specifically assesses bleeding leads to a better assessment of the role of inflammation
  • why was the bleeding on probing index not assessed? It is clearly a measure of disease activity
  • How long did the motivation phase last?- It is not clear whether full mouth periodontal treatment was employed
  • what is the scientific basis for the 0.5% HbA1c outcome?
  • Results
  • The periodontal outcomes regarding the probing depth and attachment level could have been presented differently, such as, for example, percentage of sites with PS ≥5mm or percentage of reduction of sites with PS ≥5mm or percentage of sites with attachment gain≥1 mm or number of sites with CAL≥4 mm, before and after treatment. Once these data are obtained, then perform the tests to assess correlation. The use of medians can dilute the effect of periodontal treatment and does not demonstrate the actual periodontal state in terms of the severity and extent of the disease.
  • Correct median value in item 3.4
  • Discussion
  • Some aspects should be discussed, such as the fact that the intervention group has a higher median of BMI, HbA1c and for most periodontal indexes.
  • Discuss the absence of difference between groups for HbA1c. Could it be a Hawthorne effect?
  • page 13, line 481, correct for glycated
  • from page 13 line 508 to page 14 line 511. This conclusion cannot be inferred, since both groups had a reduction in HbA1c, with no significant difference between them. What should be discussed here is the fact that the intervention group initially had a higher level of HbA1c and, therefore, the treatment effect was more significant. It is a good result for periodontal treatment.
  • page 14, the main result that is the reduction of CRP, a recognized inflammation marker, has been little discussed and explored.

Author Response

Reviewer 3:

Dear authors,  

This is an interesting article but some aspects need to be explained.

Broad and comprehensive introduction, although a little long. Some aspects should be pointed out:

Authors: The introduction has been modified, as requested. The new text is the following:

"Despite advances in prevention and treatment measures, diabetes mellitus remains one of the most important chronic diseases worldwide, associated with high mortality and morbidity, with an estimated 300 million cases in the year 2025 [1]. Research has pointed to the multiplicative effect when several risk factors are present and, of interest, scientific evidence underlines the possibility of closely linked inflammatory etiology. Accumulating evidence supports a key role of periodontal inflammation in the pathophysiology of several systemic disorders [2-4]. Progress in understanding the pathogenesis of diabetes has generated increasing interest in targeting inflammatory pathways and biomarkers of inflammation to help prevent and control diabetes and related sequelae [5-8]. Periodontitis is a chronic multifactorial inflammatory disease affecting the tissues surrounding the teeth [7], and has a prevalence estimated at 20% to 50% of the global population. Commonly, over 10% of older population have severe disease [8-10]. Periodontitis can promote systemic chronic inflammation (SCI) that can, in turn, lead to exacerbation of type 2 diabetes [9-11]. This consistent clinical evidence for an interconnection between periodontitis and diabetes risk comes from multiple randomized controlled trials (RCTs) and epidemiological investigations data that have proven that periodontal treatment resulted in HbA1c levels reduction [10,12,13]. All these findings are in harmony with the hypothesis that periodontitis and diabetes may fuel each other in a bidirectional pathway [9,13], in which a reciprocal action of inflammatory mediators leads to worsen of pathological conditions, becoming closely linked diseases [11].

The metabolic dysregulation in diabetes associated with hyperglycaemia and hyperinsulinemia are predisposing factors to infectious diseases in diabetes. Periodontitis is one of the most common complications of diabetes mellitus, affecting more of 50% of the diabetic population [28]. Periodontitis has been implicated in an increased risk of type 2 diabetes, and diabetes patients are two-five times as likely to develop risk of suffering periodontitis compared to healthy individuals, but the molecular mechanisms linking these two diseases are partially known [29,30,5] Mechanistic studies in in vivo conditions agrees to assert that among regulatory factors of this connection should be explored [31]. Specifically, pathway model that aggregates the knowledge about the impact of diabetes on periodontal status have been well-documented and characterized [28]. Data on the clinical impact of diabetes on periodontal inflammation enhancement come from two sources: the potential direct effect of hyperglycaemic status which may be responsible for the oral microbiome shifting, impaired cellular function and host defence, and upregulation of the circulant proinflammatory mediators known to be released primarily by immune cells [18,23]; the accelerated glycation of protein and lipids mediated by AGES and fatty acids which may indirectly contribute to degenerative process of periodontal tissues and influence negatively osteogenesis [30-33]. In contrast, it has been assumed that long term low-level exposure to periodontal inflammation have a negative impact on metabolic control, but the pathway by which periodontitis contributes to worsening diabetes is still unclear [30]. Human studies have demonstrated that patients suffering from periodontitis exhibit higher concentrations of pro-inflammatory cytokines- such as tumor necrosis factor-alpha (TNF-α), C-reactive protein (CRP), interleukin-6 (IL-6), Il-1β. Several mechanisms of altered macrophage polarization during obesity have been recently suggested. Based on this evidence, dysregulation in the cytokine network may contribute to development or sustaining diabetes via activation of the chronic systemic inflammation [17,28-30]. The clinical relevance of increases of CRP in subjects with periodontitis has been demonstrated, by showing that the peripheral inflammatory response is reflected in increased concentrations of C-reactive Protein in patients with periodontal infections compared to periodontally health individuals. The most reliable evidence for the co-occurrence of diabetes and periodontitis and the correlation between the two diseases is a function of inflammatory pathway [2,5,13,26].

Based on the simultaneous occurrence of periodontitis and diabetes, as recorded in most of clinical trials, this paper briefly resumes the conceptual cycle that accounts for the theory of bidirectional relationship between diabetes and periodontitis and explores the impact of suppression of periodontal inflammatory response pathway in minimising the interrelation between the peripheral and systemic inflammatory status. In this perspective, the chief object of this trial was aimed to investigate the effectiveness of non-surgical periodontal treatment on glycaemic control in patients diagnosed type 2 diabetes and, simulyaneously, the potential improvement of systemic inflammatory status."

material and methods

  • The classification of periodontal disease must be updated. According to the description, the classification of 2018 was used, not 1999.

Authors: The classification of 1999 was only cited for the brief historical description. “Diagnostic criteria of periodontitis have been validated since 1999 [23] by the American Academy of Periodontology (AAP) and have been enlarged in the 2017 revision [24]”

  • It is not mentioned if the person who performed the periodontal examinations was a trained and calibrated examiner.

Authors: The clinician was a trained and calibrated examiner. The affermation has been added, as follows: “All examinations were performed by a trained and calibrated examiner”

  • the statement that patients with insulin-dependent diabetes were excluded is mistaken, since patients with DM2 may also need insulin therapy. It must be said that patients with DM1 or other types were excluded

Authors: The insulin-dependent diabetes describes only Type 1 Diabetes. Type 2 Diabetes is also known as “non-insulin-dependent” Diabetes.  However, we have modified the description, as requested.

  • It is mentioned that smokers were excluded. what about ex-smokers?

Authors: ex-smokers were considered as smokers.

  • What is the dosage method for HbA1c? Is it certified?

Authors: The patients were recruited from the diabetic medical centre where the laboratory measurement of plasma glucose concentration was performed on venous samples with enzymatic assay techniques.

  • What is the CRP dosage method?

Authors: The patients were recruited from the diabetic medical center where the laboratory measurement of CRP concentration was performed on venous samples with enzymatic assay techniques.

  • it is not clear how the median for plaque and gingival indexes was obtained. Why was a dichotomous index not used, such as the presence or absence of biofilm and the presence or absence of marginal bleeding? the use of an index that specifically assesses bleeding leads to a better assessment of the role of inflammation

Authors: In the section “Assessment of clinical periodontal parameters” the use of Plaque Index and Ginval Index was reported, as follows:

“At baseline as well as at 3 and 6 months after non-surgical periodontal treatment, the following clinical parameters were detected to record gingival and periodontal status by a single examiner using a standardized manual periodontal probe (CP11 Hu Friedy, Europe) in six sites of each tooth (distobuccal surface, centre of vestibular surface, mesio-buccal surface, distolingual surface, centre of lingual surface and mesio-lingual surface): Pocket Depth (PD) [1], to assess the periodontal status by measuring the space between the pathologically detached gingiva and the tooth; Clinical Attachment Level (CAL), to assess the periodontal status by measuring the distance from the cemento–enamel junction of the tooth to the pocket base; Gingival Index (GI-Löe19), to assess the severity of gingival inflammation on the basis of chromatic evaluation, consistency, and bleeding on probing; Plaque Index (PI-Silness and Löe), to assess the thickness of bacterial plaque at the cervical margin of the teeth. In order to reduce the effects of the examination variability, the technique proposed by Silness and Löe was used on determining PI score [2], mesial (M) and distal (D) measurements...”

  • why was the bleeding on probing index not assessed? It is clearly a measure of disease activity

Authors: It was used the Gingival Index: “Gingival Index (GI-Löe19), to assess the severity of gingival inflammation on the basis of chromatic evaluation, consistency, and bleeding on probing;”

  • How long did the motivation phase last?- It is not clear whether full mouth periodontal treatment was employed.

Authors: The motivation is part of the periodontal treatment and is perpetuated during all phases of therapy. The full mouth periodontal treatment was employed. It has been added

  • what is the scientific basis for the 0.5% HbA1c outcome?

Authors: One of the major considered studies is the follows:  Engebretson S, Kocher T. Evidence that periodontal treatment improves diabetes outcomes: a systematic review and meta-analysis. J Clin Periodontol 2013;40(Suppl.14):153–63

  • Results
  • The periodontal outcomes regarding the probing depth and attachment level could have been presented differently, such as, for example, percentage of sites with PS ≥5mm or percentage of reduction of sites with PS ≥5mm or percentage of sites with attachment gain≥1 mm or number of sites with CAL≥4 mm, before and after treatment. Once these data are obtained, then perform the tests to assess correlation. The use of medians can dilute the effect of periodontal treatment and does not demonstrate the actual periodontal state in terms of the severity and extent of the disease.

Authors: We used the median because our data didn't meet the assumptions of the parametric test

  • Correct median value in item 3.4

Authors: Corrected

  • Discussion
  • Some aspects should be discussed, such as the fact that the intervention group has a higher median of BMI, HbA1c and for most periodontal indexes.

Authors: We are sorry, but we have registered higher medians for both the groups. This aspect was described in many points of the text.

  • Discuss the absence of difference between groups for HbA1c. Could it be a Hawthorne effect?

Authors: The Hawthorne effect refers to a phenomenon in which participants alter their behavior as a result of being part of an experiment or study. The absence of difference between groups for HbA1c cannot be explained by the Hawthorne effect. The HbA1c concentration is determinaed by several systemic factors.

            page 13, line 481, correct for glycated

Authors: Corrected

  • from page 13 line 508 to page 14 line 511. This conclusion cannot be inferred, since both groups had a reduction in HbA1c, with no significant difference between them. What should be discussed here is the fact that the intervention group initially had a higher level of HbA1c and, therefore, the treatment effect was more significant. It is a good result for periodontal treatment.

Authors: Corrected

  • page 14, the main result that is the reduction of CRP, a recognized inflammation marker, has been little discussed and explored.

Authors: Discussed, as follows: "It is generally accepted that inflammation cytokines like C-reactive protein (CRP) are determinant for the connection between diabetes and periodontitis. A meta-analysis conducted by Teeuw et al. showed a significant reduction of CRP level after periodontal treatment, as well as Katagiri et al. observed a strong relationship between the change of CRP and HbA1c concentration. Several large scale cross-sectional studies reported elevated levels of serum CRP in gingivitis and periodontitis. The effectiveness of periodontal treatment on CRP levels was also investigated from several reports indicating that CRP is consistently elevated in periodontitis individuals (>2.1 mg/l) compared with healthy controls. Our results are consistent with previous funding which reported that periodontal treatment determined a statistically significant decrease of the CRP plasma levels after 3 months. It has been documented that 3 months post treatment is a suitable interval for the primary evaluation of non-surgical periodontal treatment. Therefore, our data confirm that periodontal therapy can significantly reduce systemic inflammation by improvement of the periodontal status."